# Fueling Processes on (Sub-)kpc Scales

Francoise Combes 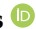

Observatoire de Paris, LERMA, Collège de France, CNRS, PSL University, Sorbonne University,
75014 Paris, France; francoise.combes@obspm.fr

**Abstract:** Since the 1970s, astronomers have struggled with the issue of how matter can be accreted to promote black-hole growth. While low-angular-momentum stars may be devoured by a black hole, they are not a sustainable source of fuel. Gas, which could potentially provide an abundant fuel source, presents another challenge due to its enormous angular momentum. While viscous torques are not significant, gas is subject to gravity torques from non-axisymmetric potentials such as bars and spirals. Primary bars can exchange angular momentum with the gas within corotation, causing it to spiral inwards until reaching the inner Lindblad resonance. An embedded nuclear bar can then take over. As the gas reaches the black hole's sphere of influence, the torque becomes negative, fueling the center. Dynamical friction also accelerates the infall of gas clouds closer to the nucleus. However, because of the Eddington limit, growing a black hole from a stellar-mass seed is a slow process. The existence of very massive black holes in the early universe remains a puzzle that could potentially be solved through direct collapse of massive clouds into black holes or super-Eddington accretion.

**Keywords:** galaxies: active nuclei; galaxies: bars; spirals; black holes; angular momentum; molecular torus; fueling; feedback; warps

## 1. Introduction

One of the main issues when attempting to explain the fueling of active galactic nuclei (AGN) is the mechanism to get rid of the angular momentum (AM). A black hole may swallow neighboring stars, which will create a depletion among those stars with a low AM. A long relaxation time will be required to replenish this loss cone, so the fueling will rely on the gas infall. The latter requires gravity torques, tangential forces, and therefore non-axisymmetric features, such as bars or spirals. Large-scale features are not sufficient and should be supported by embedded bars to prolong their action towards the center.

The high spatial resolution provided by the ALMA interferometer can reveal these embedded structures, in particular nuclear spirals inside the nuclear rings, corresponding to the inner Lindblad resonance (ILR) of the bars. Examples of these structures have been unveiled in nearby Seyfert galaxies.

Inside these nuclear spirals, at a scale of 10 pc, molecular tori have been revealed as circumnuclear disks that are kinematically decoupled from the large-scale disk. The origin of the decoupling could be as a result of several causes, like the accretion of gas with different AM, and/or precession and warping of the very central disk due to relativistic effects, coupled with the supermassive black-hole spin.

## 2. Angular Momentum Problem

This problem is as a result of the large contrast between the angular momentum of the gas in the last stable orbit, $L = 2 \times 10^{24} (M/10^8 M_\odot)$ cm$^2$/s, for a typical black-hole mass of $10^8 M_\odot$, and the gas AM at 3 kpc, $L = 10^{29}$ cm$^2$/s. The ratio of these two values means that the gas has to lose 5 orders of magnitude in AM within a relatively short dynamical time. The illustration of the AM increase with radius is shown in Figure 1. For a typical luminosity of $10^{46}$ erg/s, the central engine has to swallow 2 M$_\odot$/yr during a duty cycle of 100 Myr.

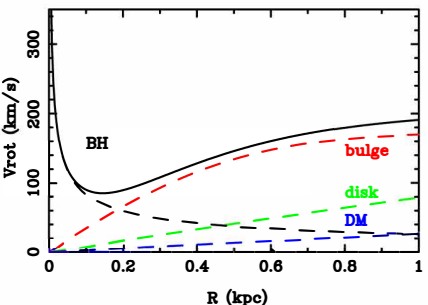 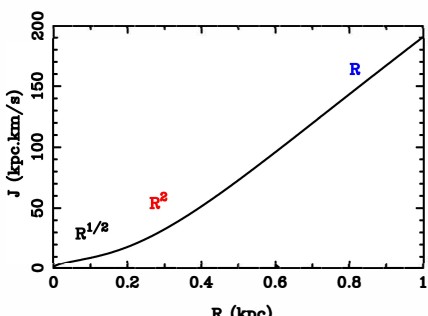

**Figure 1.** How does the angular momentum increase in radius around a super-massive black hole(BH)?: (**left**) the typical rotation curve in a spiral galaxy is dominated by the BH mass in the center (Keplerian potential), then rises due to the bulge, and is finally flat due to the conspiracy of the disk and dark matter halo (DM). These various contributions are marked in different colors; (**right**) the corresponding angular momentum per unit mass increases with the power laws of the radius, with the first slopes being a half (Keplerian), then two and one.

Stars with a low AM in the neighborhood of the black hole can be tidally sheared, and their gas is accreted in a rotating disk. These TDEs (tidal disruption events) can occur at a frequency of once every 10,000 years in the Milky Way. Some TDEs have been detected in external galaxies, signified by a characteristic light curve decreasing with time as a $-5/3$ power law [1]. However, after a brief period, the depletion and loss cone effect deplete this fueling source unless galaxy interactions re-shuffle the stellar distribution, creating nuclear star clusters in a nuclear starburst.

Gas is, however, the main fuel, and is driven inwards through non-axisymmetries in the galaxy potential. Several steps can be distinguished in this process. First, primary bars, typically with diameters of 10 kpc, drive the gas from their corotation to R~100 pc, where the gas is stalled in a ring.

Then, embedded nuclear bars prolong their action from 100 pc to 10 pc. Non-axisymmetries usually have morphologies of m = 2, but m = 1 (lopsidedness), or tidal forces from companions can also play a role.

At smaller scales of 1–10 pc, other processes also have to be considered, such as turbulence, viscosity, warps, bends, dynamical friction, and the formation of thick disks, as long as a sufficient amount of gas remains.

### 2.1. Dynamics of Bars

Let us recall the main features of barred galaxies: the stellar orbits are classified through a skeleton of periodic orbits. The latter are orbits that close on themselves after one or more turns in the bar rotating frame. These are the building blocks that determine the stellar distribution function, as they define families of trapped orbits around them. Trapped orbits are non-periodic, but oscillate about one periodic orbit, with a similar shape. The periodic orbits are numerous (see the review by Contopoulos and Grosbol [2]); we will discuss the most important ones for bar support in the following. Inside corotation, the x1 family is the main family supporting the bar. Orbits are elongated parallel to the bar, within the corotation. The x2 family also exists, but only between the two inner Lindblad resonances (ILRs), if they exist. They are more round and are elongated perpendicular to the bar. The existence of two ILRs in the axisymmetric sense might not be sufficient for the x2 family to appear. When the bar is strong enough, the x2 orbits disappear. The bar strength necessary to eliminate the x2 family depends on the pattern speed; the lower the speed, the stronger the bar must be. Outside the corotation, the 2/1 orbits that are run in the retrograde sense in the rotating frame are perpendicular to the bar inside the outer Lindblad resonance (OLR), and are parallel to the bar slightly outside (see discussion by Kalnajs [3]). Their shape is a characteristic figure eight, which is very similar to the dimpled shape of some outer rings in barred galaxies.

In summary, periodic orbits are aligned parallel or perpendicular to the bar, and their orientation changes by 90° at each resonance [4,5]. Gas tends to follow periodic orbits, but its dissipative character, due to cloud collisions, means that orbits cannot cross. Instead, their orbits are tilted, and they change gradually by 90° at each resonance. The crowding of these stream lines produces a spiral morphology [6]. The spiral is open, and at its maximum can rotate by 180–360°.

### 2.2. Embedded Structures

Because the gas and stars are not in phase, stellar bars exert torque on the gas, except at resonances where the gas piles up and stalls in rings aligned with the bar in some way. When mass has accumulated in the center, all frequencies of $\Omega$ and $\Omega - \kappa/2$, the orbit precessing rate, increase significantly in value, implying the existence of two ILRs. Between these ILRs, the periodic orbits (x2) are perpendicular to the bar.Stars cannot sustain the bar anymore and weaken it. The z-resonance, creating peanut-shape bulges, also weakens the bar. This triggers the decoupling of a second bar, an embedded nuclear bar [7]. This second bar rotates faster than the primary bar, and both are misaligned [8].

### 2.3. Fueling AGN: Removing Angular Momentum (AM)

The torques exerted on the gas by the bar change sign at each resonance (cf Figure 2). Outside the corotation radius (CR), the gas is driven outwards and accumulates at the outer Lindblad resonance (OLR). Inside CR, the gas is driven inwards, at least down to ILR. To quantify the phenomenon, the bar gravity torque can be computed from the red images by tracing old stars, and using the potential. When the gas distribution is overlaid on the torque map (cf Figure 3), the sign and amount of the gas infall can be estimated, as in NGC3627 [9]. The correlation between the bars and AGN is still debated [10–13], as the time-scales corresponding to the primary and secondary bars are quite different, as well as being different from the AGN duty cycle.

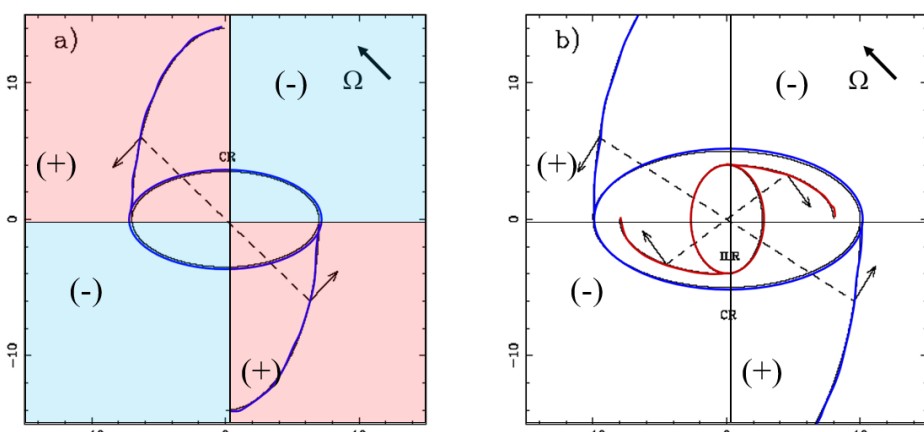

**Figure 2.** The sign of the torques exerted by the bar on the gas can be obtained geometrically through these schematic diagrams: (**a**) outside corotation, or roughly outside the bar, the torque is positive (red quadrant) with respect to the sense of rotation ($\Omega$), and the gas is driven out toward the outer Lindblad resonance (OLR); (**b**) inside the corotation, the torque is negative (blue quadrant), and the gas is driven inward, down to the inner resonance (ILR).

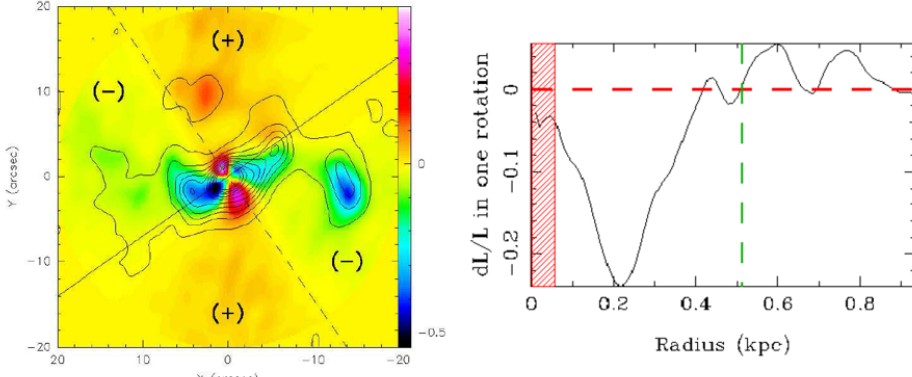

**Figure 3.** The gravity torques can be measured on real galaxies by computing the potential in each pixel and the forces from a red image (old stars), and comparing the results with the observed gas density. The (**left**) image is the torque map in color, with the bar splitting the plane in four quadrants, for NGC 3627. The gas density is overlaid in contours. The (**right**) plot quantifies the relative angular momentum lost in one rotation, while averaging the torque over the azimuth, weighted by the gas density. Adapted from Casasola et al. [9].

During a survey of about 20 galaxies using the IRAM interferometer, statistics of fueling at the 10–100 pc scale were obtained [14]. Only 35% of negative torques were measured in the center; the remainder of the measurements were positive torque or gas was stalled in a ring. For the latter case, future fueling has to wait for the decoupling of a secondary bar, as shown in the simulation of Figure 4. This means that the fueling phases are short, a few $10^7$ years, and may be due to feedback. Star formation is also fueled by torque, and is always associated with AGN activity as well as longer time scales.

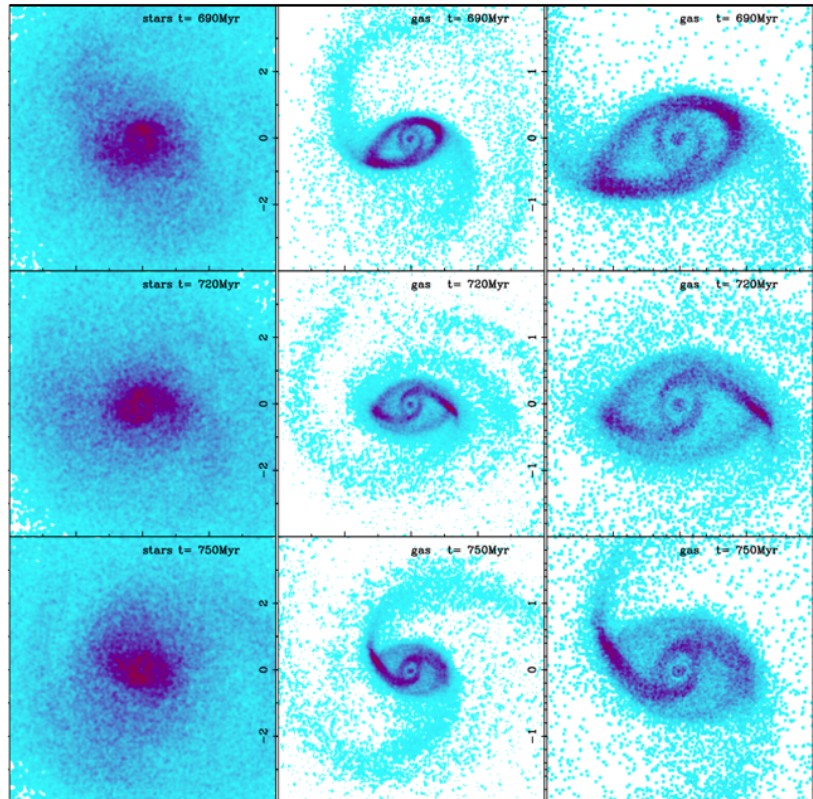

**Figure 4.** Hydro-N-body simulation of a double bar in a spiral galaxy. The (**left**) panel shows the stellar component surface density (linear scale, axes in kpc), the (**middle**) panel shows the gas

component (log scale), while the (**right**) panel is also the gas but in a zoomed spatial scale. The simulation shows clearly that the gas first piles up in the external ring (ILR of the primary bar), then progressively infalls inside the ILR to gather at the nuclear bar resonance. Adapted from Hunt et al. [15].

## 3. Small-Scale Fueling with ALMA

With the advent of ALMA, higher spatial resolution is possible, and nearby galaxies can be explored using 10 pc scales. One of the first barred spiral observed was NGC 1433 (cf Figure 5). While only a star-forming ring was observed at ILR with HST, a second ring was detected in the molecular gas with ALMA, corresponding to the second ILR. The computation of the torques showed that AGN is not presently fueled, but positive torques bring the gas from the center to the second ring. Negative torques outside this second ring contribute to the accumulation of gas there [16]. On the minor axis, an outflow with small velocity has been detected. It is one of the smallest outflows that has been detected, with $M(H_2) = 3.6 \times 10^6$ $M_\odot$ and a flow rate of 7 $M_\odot$/yr [17].

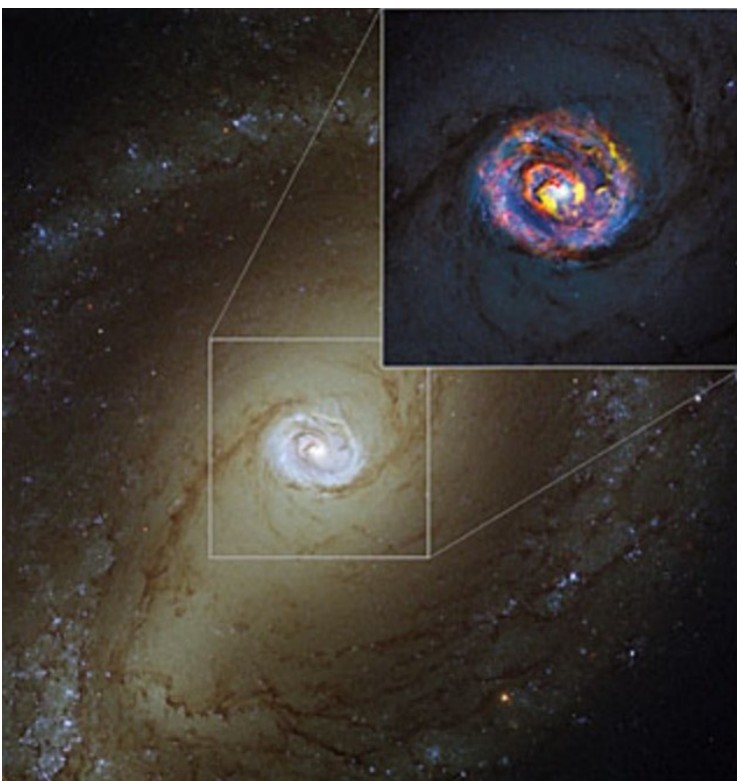

**Figure 5.** HST image of NGC1433 bar, with characteristic leading dust lanes, resulting in a blue ring at the inner Lindblad resonance (ILR). The insert displays the blue ring from HST overlaid with the orange-red image from ALMA of the molecular gas, traced by its CO(3-2) emission. The latter reveals a second ring inside the first ILR. Adapted from Combes et al. [17].

Other barred Seyfert galaxies, like NGC1566, were found in the feeding phase (cf Figure 6). Inside the ILR ring, the CO(3-2) emission map from ALMA revealed the existence of a nuclear disk, with a trailing nuclear bar. The existence of the trailing spiral was a surprise, as without the gravitational influence of the central black hole, a leading spiral was expected. This was due to the shape of the $\Omega - \kappa/2$ curve as a function of the radius. When the gas would infall, it precessed more rapidly if the curve increased when the radius was reduced, which means a trailing arm; the inverse was true if the curve declined (cf Figure 7).

If the spiral was trailing, this means that the gas experienced a climbing curve, due to BH, and it entered the sphere of influence of BH. The computation of the torques in NGC1566 confirmed the evidence of fueling [18].

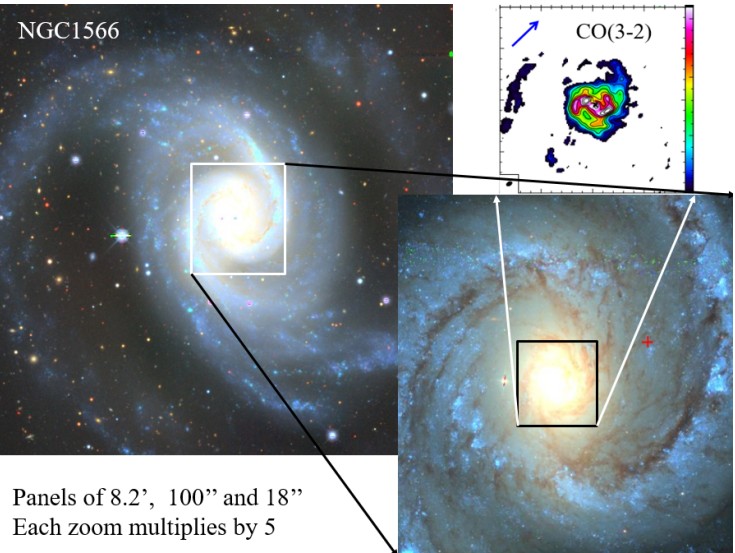

Panels of 8.2', 100'' and 18''
Each zoom multiplies by 5

**Figure 6.** The barred spiral galaxy NGC 1566 reveals several embedded structures, as seen in these three progressively zoomed images. The last one reveals a trailing nuclear spiral in the molecular gas, traced by its CO(3-2) emission observed with ALMA. Adapted from Combes et al. [18].

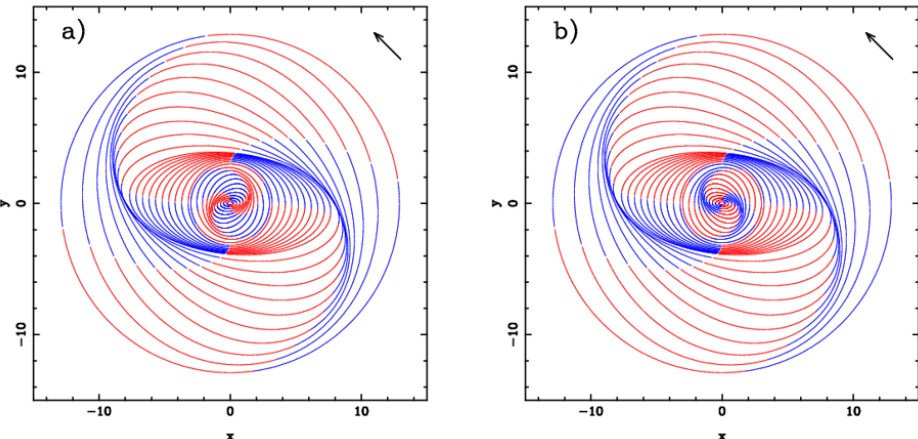

**Figure 7.** Gas streamlines schematically represented by elliptical orbits, precessed to form logarithmic spirals. The streamlines are initially aligned on the horizontal axis (parallel to the bar) and colored according to the four quadrants. The pattern speed is such that an ILR exists inside the bar, delineated by a ring. (**a**) without a super-massive black hole (BH), the precessing rate decreases towards the center and the gas forms a leading nuclear spiral. (**b**) with a BH dominating the potential inside the ILR, the precessing rate increases towards the center and the gas forms a trailing spiral. This changes the sign of the torque exerted on the gas.

In several other barred galaxies, a trailing nuclear spiral has been revealed in the molecular component with ALMA, for instance in NGC 1808 or NGC 613. This indicates that the gas has entered the sphere of influence of the black hole, making the torque negative, ensuring the AGN is fueled. This trailing spiral always develops inside the ILR ring of the bar [19].

NGC 613 is the academic case of a strong barred galaxy, with a star-forming gas ring at ILR. The first ALMA observations, with a moderate spatial resolution, could see only the ring, without resolving the internal structure [20]. With a beam of 60mas, it was possible

to clearly see an internal trailing nuclear spiral and a molecular torus inside the spiral (cf Figure 8). The computation of torques indicated that, when inside a 50 pc radius, the gas could lose all its angular momentum in only one rotation [21]. In addition, a molecular outflow was detected along the minor axis, parallel to the cm wave detected by the radio jet.

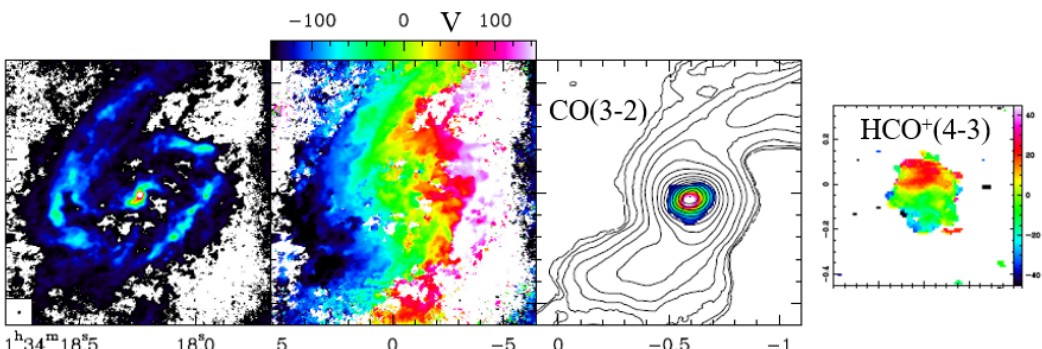

**Figure 8.** The barred galaxy NGC 613 shows a contrasted ring at its ILR, in the molecular gas traced by CO(3-2) with ALMA. From left to right is the gas surface density, then the velocity field, and a 10-fold zoom of the gas density (contours), with the radio continuum (color-scale) at the center. The right-most panel is the velocity field of the HCO$^+$(4-3) emission, revealing a misalignment with the large-scale gas. Adapted from Combes et al. [22] and Audibert et al. [21].

## 4. Molecular Tori

Our vision of the central regions of AGN surrounding the black hole have changed significantly in recent years. For a long time, a dusty torus with a donut shape was assumed to exist around the accretion disk and the broad line region (BLR). It was thought to obscure the BLR for observers observing the accretion disk almost edge-on. But infrared interferometers have shown that the dust is frequently detected in the polar direction, instead of being aligned along the putative torus. This polar dust must be ejected through the AGN wind, starting from the dust sublimation radius (pc size), and forming the border of a hollow cone [23]. The gas motion is then both infall in a thin disk, forming a molecular torus, and then outflows in the perpendicular direction.

Figure 8 shows that the molecular torus, inserted in the nuclear spiral, has a decoupled kinematics, i.e., the kinematic major axis is not aligned with that of the large-scale disk [21,22].

This misalignment is frequent in all nearby galaxies observed using ALMA with 10 pc resolution, such as NGC 1672 or NGC 1326 in the NUGA sample [22], or NGC 5643 or NGC 6300 in the GATOS sample [24]. It is not yet possible to determine whether the central molecular torus is simply continuously warped or tilted, with discontinuous disks torn into a few pieces. Circinus is one of the nearest galaxies, observed down to 2 pc resolution, and shows a nuclear spiral of 20–30 pc and another circum-nuclear disk inside, which can be interpreted as the molecular torus. It is self-absorbed in CO(3-2) but not in CO(6-5), nor in the dense gas tracers, such as HCO$^+$. In addition to this cold thin disk fueling the AGN, there is an outflow in the polar direction, composed of warm dust and ionized gas (cf Figure 9) [25,26].

The prototypical Seyfert 2 NGC 1068 has been intensively observed at a high resolution at many wavelengths. While the near-infrared clearly reveals a hollow polar cone in warm dust [27], the cold component in the millimeter domain reveals a molecular torus of 7–10 pc in diameter [28,29]. The torus is quasi edge-on, misaligned with the large-scale disk, warped, more inclined than the water maser disk, and might suffer from counter rotation [29,30]. In contrast, the barred Seyfert 1 galaxy NGC 1097, using ALMA at 10 pc resolution, does not show any dense and compact torus. There is a nuclear spiral inside the star-forming ring at the ILR of the bar [31,32]. The absence of torus concerns about 10–20% of low-luminosity AGN. This percentage increases with the Eddington ratio, and could be related to the AGN feedback [24].

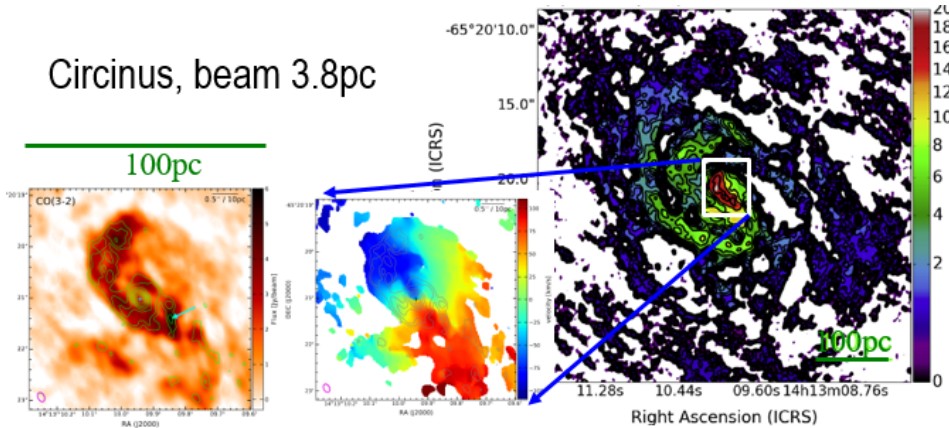

**Figure 9.** Image of the molecular gas, obtained in CO(3-2) with ALMA, of the Circinus galaxy, with a resolution of 3.8 pc. There is a nuclear spiral, of 20–30 pc, and another circum-nuclear disk inside, which could be the molecular torus (edge-on, for this type 2 AGN). Adapted from Tristram et al. [26].

The influence of the AGN activity is indeed visible on the molecular gas concentration. The latter can be quantified by the surface density ratio of the molecular gas inside 50 pc and inside 200 pc. The most active AGNs have less $H_2$ concentration. The latter drops for X-ray luminosities $Lx > 10^{42.5}$ erg/s, or for Eddington ratios $> 10^{-3}$, see Figure 10, [24].

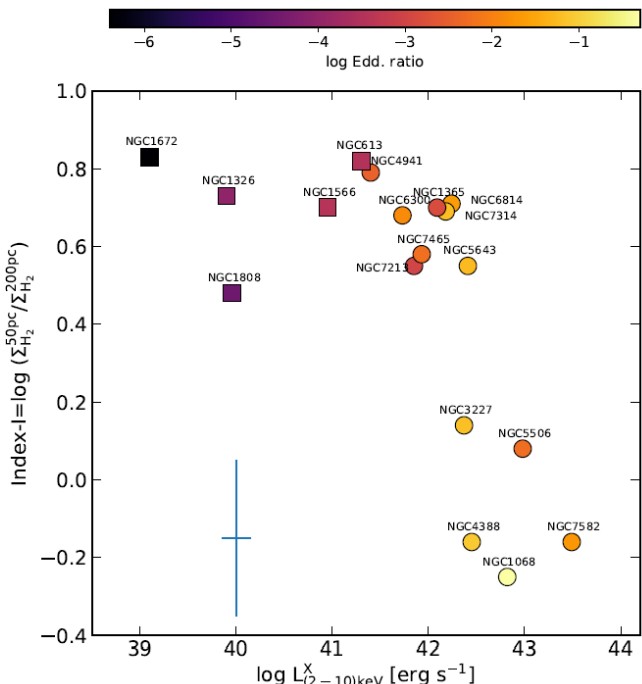

**Figure 10.** Concentration of molecular gas in the central regions of galaxies versus their AGN luminosities measured in the 2–10 keV X-ray band. The concentration is measured using the ratio of the average $H_2$ surface densities at two spatial scales, $r < 50$ pc and $r < 200$ pc, characteristic of the nuclear and circumnuclear regions. Symbols are color-coded as a function of the Eddington ratios. The sample galaxies: NUGA (square markers) and GATOS (circle markers) can be separated both by their AGN activity and molecular gas concentration. Adapted from García-Burillo et al. [24].

## 5. Early Black Holes

Along their lifetime, galaxies grow both their bulge and their black hole, in synergy, so that a tight relation can be observed between both masses [33]. Secular evolution, together with some tidal interaction with companions, can provide the fuel for AGN and BH growth. However, AGN feedback makes the BH growth intermittent, as described above. Some

different processes should occur at a high redshift, to account for the observations of high black-hole masses ($M > 10^9$ $M_\odot$) already at $z > 6$ [34]. Hundreds of quasars have been detected at a high redshift, $z > 6$, with black hole masses between $10^8$ and $10^{10}$ $M_\odot$, in the first billion years of the universe (e.g., [35–37]). The black-hole growth rate through accretion is proportional to its mass, and is thus very slow at the beginning, starting with stellar mass seeds. At any time, the Eddington luminosity imposes an upper limit in the accretion rate, which is proportional to the BH mass. In contrast, at a high redshift, black holes appear to grow faster than their host bulges, and the proportionality relation breaks. Several solutions have been proposed. One is to assume that the BH in the initial phase accrete exceptionally faster than their Eddington rate, without being stopped by the AGN feedback.

Simulations of black-hole growth with super-Eddington accretion, even in a massive over-density environment, progenitor of a cluster, in general, do not reproduce the observed massive black holes at $z = 6$ [38], nor explain why black-hole growth is more rapid than the stellar mass growth, contrary to what is observed at $z = 0$. However, with some modifications to the usual scenario, taking into account less efficient AGN feedback, more efficient accretion, and starting earlier with a more massive seed, it is possible to account for the observations [39].

An alternative solution is to assume that BH collapses directly from a massive cloud, due to primordial metal abundance and a lack of fragmentation. The seeds forming the first black holes would then be much more massive than stellar, and the growth rate would be strong from the start. This, however, assumes contrived suppression of the $H_2$ molecule formation, and remains debated.

The existence of intermediate mass black holes (IMBHs) is part of the puzzle, as they have not yet been detected unambiguously. The direct collapse scenario could explain their low abundance. The major challenge for observations of IMBHs is their small mass and small impact on their surroundings. As the radius of influence of a black-hole is scaled linearly with its mass, it requires a very high spatial resolution in order to detect their impact on the gas or stellar component. Also, the dynamical friction time-scale for a black hole to decay towards the nuclei is inversely proportional to the mass, and IMBH might not have time to infall within a Hubble time (e.g., [40]).

An promising way to detect IMBH is to search for AGN signatures, and candidates have been observed in several low-mass galaxies [41]. Candidates for IMBH exist in several dwarf active galaxy nuclei (e.g., [42]), however, in the core of globular clusters, where they were expected, only a collection of stellar-mass black holes have been detected instead [43,44].

## 6. Summary

AGN fueling requires non-axisymmetries in the galaxy potential to drive the gas inwards. The primary bar in spiral galaxies can drive the gas from its corotation to the inner Linblad resonance, where the gas accumulates and forms a starbursting ring, at the scale of 100 pc. Further fueling has to await the formation of a nuclear bar, embedded inside the ILR ring. ALMA has revealed 10 pc scale structures inside the ring, in particular, trailing nuclear spirals, indicating that the gas has entered the sphere of influence of the central black hole. The stellar potential then prolongs the action of the primary bar to fuel the AGN. Inside the nuclear spiral, a molecular disk, kinematically decoupled, is often found, which can be interpreted as the molecular torus.

The misalignment of the torus can be explained by gas accretion with a different angular momentum. This occurs naturally through star formation/supernovae feedback, which ejects gas above the plane. When this gas comes back, by the fountain effect, it can arrive in any direction with a random angular momentum. Numerical simulations have shown that even polar gas rings or disks can be formed [45,46].

The decoupling of molecular tori is not too surprising, given the very different dynamical time scales between the 10 pc and 100 pc scales. In addition, the material at much less

than 1 pc from the black hole is certainly influenced by relativistic effects, related to the black-hole spin: the Bardeen–Petterson effect, where the torque exerted by the BH tends to align the material perpendicular to its spin, and produces warps. Warping might also be radiation-driven [47,48].

Contrary to the existence of a tight relation between bulge and black hole masses in nearby galaxies, super-massive black holes grow relatively faster than the stellar component in early galaxies, in the first billion years of the universe. To reproduce this behavior, simulations have assumed super-Eddington accretion and low-efficiency AGN feedback at these epochs. Alternatively, direct collapse of super-massive stars into black holes might contribute to the solution. Future observations in the early universe with JWST might enlighten the issue.

**Funding:** This research received no external funding.

**Informed Consent Statement:** Not applicable.

**Acknowledgments:** Thanks to Ilaria Ruffa and the organisers of the conference "AGN on the beach" in Tropea, Italy, where this material was discussed. FC has benefited from the support of the Programme National Cosmologie et Galaxies.

**Conflicts of Interest:** The author declares no conflict of interest.

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
