# Peer review of "Fueling Processes on (Sub-)kpc Scales"

_galaxies, doi:10.3390/galaxies11060120_

Round 1

Reviewer 1 Report

Comments and Suggestions for Authors

I found the manuscript very interesting and well presented. I do not have any significant comment, and I believe that it should be accepted for publication as it is. A couple of typos I spotted:  

line 40: momenteum --> momentum

line 188: strucrures --> structures

Author Response

I have corrected the typos

Reviewer 2 Report

Comments and Suggestions for Authors

A very nice review. Two minor suggestions:

1. Some of the figures are of a not very high quality or straggle from small font sizes. Please improve, if possible. I do appreciate that some figures are adapted from other papers.

2. I believe, it would be useful to discuss the problem of the very massive black holes in the early Universe in the summary, elaborating on the next steps needed to resolve it.

Author Response

I have tried to improve the figures, and I have developped the

very massive black hole section

Reviewer 3 Report

Comments and Suggestions for Authors

This is an interesting review on the fuelling processes onto AGN, the recent progresses and the open questions. The paper reads well, although it may be too specialistic in some parts, where some more background information would have been useful for the non-expert. However, I think this would be outside the scopes of this review.
I do not have any particular comment, apart from very minor points:

-Figure 2 is cited after Fig. 3 and 4.

-The axes in Fig. 2 are not readable and possibly not consistent with what written in the caption

-There is a limited number of typos that can be easily spotted with an automatic spell check.

Author Response

More text has been added to explain the bar dynamics

All figures are cited in order, and typos have been corrected

Reviewer 4 Report

Comments and Suggestions for Authors

REPORT

-------

The paper "Fuelling processes on (sub-)kpc scales" by Francoise Combes (galaxies-2727358 v1) presents a review of the diverse processes that act as fueling methods for AGNs at different scales.

This review is interesting and deserves publication after only minor modifications. 

My suggestions therefore aim at making this review easier to read for the non experts and to correct minor typos and the like. 

General:

- the call to the figures in the text is out of order (esp for fig 2)

- References must be numbered in order of appearance in the text, not in alphabetical order 

  ( see https://www.mdpi.com/journal/galaxies/instructions#preparation )

Other minor issues

- line 1: main issue -> main issues

- Figure 1 caption:

   - the first sentence syntax points at a question, but with no question mark. I would suggest  either putting the question mark after '(BH)' or (my preference) remove the  'does' on the first line.

   - dominated by the BH in the center -> dominated by the BH mass in the center (?)

   - perhaps an alternative to 'conspiracy' would be synergy or  cooperation? 

   - marked in color -> marked in differnt colors 

   - (keplerian) than -> (Keplerian) then 

- Paragraps lines 46-58: perhaps the non-initiated here could benefit of a short introduction to loss cone effect and morphology, or a general reference can be given, so the reader can look it up. 

- line 59: other processes have to be considered in addition -> other processes also have to be considered

- Fig 2 caption: somewhere I would suggest stating that the different rows correspond to different,  subsequent epochs/ages.

- line 65: orientation spelt wrong

- line 68: produce -> produces

- line 81: corotation -> corotation radius

- line 82: resonance spelt wrong

- line 86: please define what the timescales are you are referring to, whether for formation, or existence.

- line 93: may -> and may

- line 94: associated to -> associated with

- line 104: - outflow -> outflows

            - 3.6 10^6 -> 3.6 x 10^6

- Fig 6: "Each"  is underlined in red

- line 143: teared -> torn

- line 144: galaxy -> galaxies

- line 151: multi wavelength -> many wavelengths

- line 154: may be suffers -> maybe suffers from

- Figure 10: - what is the blue point? Representative error bars?

             - caption: Eddington ratios: please clearly state the direction  of increasing Edd ratio with color. 

-line 191: Inside the nuclear spiral .. decoupled -> inside the nuclear spiral a molecular disk, kinematically decoupled is often found. 

- line 200: 1 pc of the black hole -> 1 pc from the black hole

Missing compulsory information at the end of the paper:

1) Author contribution: as sole author, this seems redundant, but a simple statement should be added. 

2) Funding (a funding statement is actually found in the Aknowledgements)

3) Data availability statement: one can just say that no new data were presented. 

4) conflict of interest: if none exists, a simple 'The author has no conflicts of interest' should do.  

Comments on the Quality of English Language

Minor issues detected, and noted in my report. 

Author Response

Figures are cited in rder,

I much prefer my way of citing the references (liek usual A&A journals)

Other ordering is optional

I hae corrected all language editions and typos